# *Dittrichia* *graveolens* (L.) Greuter, a Rapidly Spreading Invasive Plant: Chemistry and Bioactivity

**DOI:** 10.3390/molecules27030895

**Published:** 2022-01-28

**Authors:** Maria Ponticelli, Ludovica Lela, Daniela Russo, Immacolata Faraone, Chiara Sinisgalli, Mayssa Ben Mustapha, Germana Esposito, Hichem Ben Jannet, Valeria Costantino, Luigi Milella

**Affiliations:** 1Department of Science, University of Basilicata, Viale dell’Ateneo Lucano 10, 85100 Potenza, Italy; maria.ponticelli@unibas.it (M.P.); ludovica.lela@unibas.it (L.L.); daniela.russo@unibas.it (D.R.); immacolata.faraone@unibas.it (I.F.); chiara.sinisgalli@unibas.it (C.S.); 2Spinoff BioActiPlant s.r.l., Viale dell’Ateneo Lucano 10, 85100 Potenza, Italy; 3Laboratory of Heterocyclic Chemistry, Natural Products and Reactivity, Medicinal Chemistry and Natural Products (LR11ES39), Faculty of Sciences of Monastir, University of Monastir, Monastir 5000, Tunisia; ben.mustapha.mayssa@gmail.com (M.B.M.); hichem.bjannet@gmail.com (H.B.J.); 4Department of Pharmacy, University of Naples Federico II, 80131 Napoli, Italy; germana.esposito@unina.it

**Keywords:** *Dittrichia graveolens* (L.) Greuter, *Inula graveolens* L., stinkwort, Asteraceae, phenolic compounds, borneol, terpenes, flavonoids, invasive species

## Abstract

*Dittrichia graveolens* L. Greuter belonging to the Asteraceae family, is an aromatic herbaceous plant native to the Mediterranean region. This plant species has been extensively studied for its biological activities, including antioxidant, antitumor, antimicrobial, antifungal, anti-inflammatory, anticholinesterase, and antityrosinase, and for its peculiar metabolic profile. In particular, bioactivities are related to terpenes and flavonoids metabolites, such as borneol (**40**), tomentosin (**189**), inuviscolide (**204**). However, *D. graveolens* is also well known for causing health problems both in animals and humans. Moreover, the species is currently undergoing a dramatic northward expansion of its native range related to climate change, now including North Europe, California, and Australia. This review represents an updated overview of the 52 literature papers published in Scopus and PubMed dealing with expansion, chemistry (262 different compounds), pharmacological effects, and toxicology of *D.* *graveolens* up to October 2021. The review is intended to boost further studies to determine the molecular pathways involved in the observed activities, bioavailability, and clinical studies to explore new potential applications.

## 1. Introduction

*Dittrichia graveolens* (L.) Greuter, (common name *Stinkwort* syn. *Inula graveolens* L. Desf.), is a Mediterranean native plant. The genus *Dittrichia* belongs to the family of Asteraceae (order Asterales) and includes five species native to the Mediterranean basin [1], two of which are currently widespread in other regions [2,3,4]. *Dittrichia* reaches a height of 50 cm [5]; it is an annual plant and has a strong aroma with a camphor-like smell.

Coinciding with recent climate change in central Europe, the species is currently undergoing a dramatic northward expansion of its native range which now includes the UK, the Netherlands, and Poland. This range expansion may in part have been promoted by the rapid evolution of earlier flowering time in northern populations over the past few decades. Outside its native range, *D. graveolens* has a long history of colonizing regions with a Mediterranean climate on other continents, including Australia, South Africa, California, and now Chile. Due to their adaptation to disturbed open habitats as well as their efficient seed production and dispersal [6], species of *Dittrichia* are considered invasive in California [3,7], Australia [2], and recently colonized areas of Europe [8].

*Dittrichia graveolens* is widely used in traditional and modern medicine for its antifungal, antibacterial, anti-inflammatory, insecticide, and sedative properties. It is also known to be poisonous to livestock and causes allergic contact dermatitis in humans. *Dittrichia* is not palatable to animals, even if its aerial parts are used in Crete as a component of an external application to treat lice in chicken [9].

This herbal plant is found primarily along roadsides and grows well in disturbed upland and wetland sites on a variety of substrates, including soils with heavy metals. In California, *D. graveolens* is most common on roadsides, right-of-ways, gravel mines, detention basins, riparian floodplains, and seasonal wetlands.

Besides the negative effects that this introduction might have on local biodiversity, the species could be dangerous if ingested by animals and a potential hazard for humans.

Due to the potential of this plant in drug discovery, this study wishes to review all metabolites and biological activities reported in the literature until October 2021 and point out the attention of climate change’s spreading.

The bubble map (Figure 1) created with VOSviewer software, version 1.6.17 (© 2022, Centre for Science and Technology Studies, Leiden University, Leiden, The Netherlands) for Windows, is intended to offer a rapid visualization of the interdisciplinary work involved in this review to boost novel projects related to *D. graveolens*.

## 2. Phytochemistry

*D. graveolens* is a strongly aromatic species, and its essential oil, known as “*odorous Inula*” or “*odorous Dittrichia*”, is widely used in phytotherapy or industries in the production of perfumes, soaps, and toiletries [10,11]. Moreover, numerous studies have investigated the essential oil chemical composition (Appendix A) [11]. In particular, until now, thin-layer chromatography (TLC) [12], fractionation with Sephadex and silica gel columns [13,14,15,16,17], nuclear magnetic resonance (NMR) [18], electron ionization mass spectroscopy (EIMS) [17], high-performance liquid chromatography (HPLC) [11], and associated techniques, like gas chromatography-mass spectrometer (GC-MS) [10,19,20,21,22,23,24,25,26,27,28,29,30,31], liquid chromatography-mass spectrometry (LC-MS) [32], and gas chromatograph-flame ionization detection (GC-FID) [28] have been used to investigate the active molecules of *D. graveolens*. These methods have been applied after extracting the vegetal material, which was principally made using distillation, Soxhlet, maceration, microwave, ultrasound-assisted extraction, and supercritical fluid extraction techniques [11,25,31,33] (Figure 2).

### Chemical Characterization of D. graveolens

The essential oil of *D. graveolens* has been extensively studied from a chemical point of view. Bicyclic monoterpenes, i.e., borneol (**40**) and bornyl acetate (**41**) have been shown to be the principal metabolites of essential oil, followed by monoterpenes, camphene (**47**), terpineol (**95**) and sesquiterpenes, caryophyllene (**130**), cadinol (**114**), and farnesol (**155**) [10,19,23,24,27,29,30]. The most abundant metabolites produced by this species are oxygenated monoterpenes, followed by monoterpene hydrocarbons, oxygenated sesquiterpenes, and sesquiterpene hydrocarbons [21,29].

In 2004, Blanc et al. [19] reported using an acid-basic methodology to obtain the neutral part of the commercial oil, leading to the identification of 37 monoterpenes, 34 sesquiterpenes, and only 15 acyclic non-terpenic compounds.

It was observed that the different geographic distribution of *D. graveolens* and the harvesting season affected phytochemical composition [10], ascribing to the distinct climatic pattern of the samples [23]. Bornyl acetate (**41**) appeared to be the major compound found in *D. graveolens* essential oil. As an example, the essential oil of *D. graveolens* from a species collected in France, analyzed by GC-MS [22], was found to be constituted by 54% of bornyl acetate (**41**) followed by borneol (**40**) and camphene (**47**), 20% and 4.9%, respectively. The same results have been reported by Blanc and coworkers, in 2004, from the essential oil obtained from Corsica species collected at the full flowering stage [19]. Bornyl acetate (**41**) with a lower percentage (25.4%) was also present in the essential oil of the wild-growing Greek species collected at the full-flowering stage and investigated by Petropoulou et al., together with *epi-α*-cadinol (**116**) (30.2%) [29]. Bornyl acetate (**41**) (21.7%), followed by borneol (**40**) (18.7%), was also dominant in the essential oil of aerial parts collected from the Stara Planina mountain in Serbia [26]. On the other hand, borneol (**40**) was prevalent in the oil obtained from the Iranian species with a percentage of 60.7% [30], but only with 12.8% in the Greek essential oil [26,29]. Instead, it was present in an intermediate percentage (43.6%) in the essential oil of Montenegrin *D. graveolens*, followed by bornyl acetate (**41**), caryophyllene oxide (**128**), trans-*p*-mentha-1(7),8-dien-2-ol (**77**) and dehydro-1,8-cineol (**56**) (38.3%, 2.5%, 2.2%, and 1.2%, respectively) [28]. The hydrodistilled of the fresh aerial parts of *D. graveolens* collected in Constantine (North Easter Algerian) was shown to contain the iso bornyl acetate (**42**), instead of bornyl acetate (**41**), in a high percentage (50.8%), followed by borneol (**40**) (18.3%) and *τ*-cadinol (**116**) (6.2%) [20]. Interestingly, the essential oils obtained from *D. graveolens* collected in Morocco, Algeria, Iran, Serbia, Lebanon, Turkey, and Corsica have oxygenated monoterpenes as main compounds, and, in particular, the bornyl acetate (**41**) and borneol (**40**) [19,20,21,23,26,28]. The highest percentage of oxygenated compounds was reported in *D. graveolens* oil from Lebanon and Turkish origin plants [23].

Differently, the major compounds found in the hydrodistilled essential oil obtained from Iranian aerial parts was 1,8-cineole (**55**) followed by *p*-cymene (**61**) (54.89% and 16.2%, respectively) [10]; while selin-11-en-4-*α*-ol (**185**), 1,10-di-*epi*-cubenol (**144**), and cedr-8(15)-en-9-*α*-ol (**134**) (14.1, 10.3 and 10.3%, respectively) were the main compounds of hydrodistilled essential oil obtained from the flowering species in Monserrato (southern Sardinia) [25].

Different results were obtained when the same flowering species was subjected to supercritical CO_2_ extraction (SFE). In fact, the three main compounds, present in the hydrodistillate, were present in the SFE extract in a lower percentage (respectively 3.5, 5.7, and 9.7% for selin-11-en-4-*α*-ol (**185**), 1,10-di-*epi*-cubenol (**144**), and cedr-8(15)-en-9-*α*-ol (**134**)). At the same time, the main compound was found to be caryophyllene oxide (**128**) (14.3%), followed by cedr-8(15)-en-9-*α*-ol (**134**). Bornyl acetate (**41**) and borneol (**40**) were not even present in trace amounts [25].

A further investigation on the chemical composition of the essential oil was carried out by Sellem et al. in 2020 [30] by comparing the variation of the chemical composition related to the seasons in which the plant has been collected. In particular, the essential oil was produced starting from *D. graveolens* collected at Chebba salt marsh in the months of April, July, October, and January. The differences found mainly concern the odor and the quantitative of metabolites rather than the qualitative composition of the oil. The highest oil quantitative was obtained from the plants harvested in July (0.678%). In addition, this oil had a stronger odor than those obtained in the other three seasons. Authors justify these results by observing that *D. graveolens* in summer is in the flowering time and therefore at the best of its activity in attracting pollinating insects. Moreover, the season influences the composition of oil and the percentage of metabolites. The oil obtained in the summer showed the highest content in bornyl acetate (**41**), borneol (**40**), and thymol (**99**) that drastically decreased in the other seasons [30]. As previously stated, in 2004, Blanc et al. [19] reported similar results. On the other hand, *β*-selinene (**186**) and manool (**179**) were identified only in the oil obtained in July. In contrast, camphene (**47**) and 1,8-cineole (**55**), reached their lowest value in July. In addition, *τ*-cadinol (**116**), *α*-terpineol (**95**), carveol (**52**), and *β*-caryophyllene (**129**) recorded the high amount in October and then dropped dramatically. Among factors that influence the different chemical composition of the essential oil, there are the brightness, which is different in the four seasons, and the thermoregulation; in fact, the hydrophobic compounds protect the plant from drying out and therefore increase in properties, thus reaching larger quantities [30].

The hydrodistillation process used to produce the essential oil gave an aqueous residue. Its release could represent a risk of environmental pollution. For these reasons, Gharred et al. investigated the aqueous residue of dried and ground leaves and flowers hydrodistillation by HPLC analysis identifying flavonoid compounds. The authors identified quercetin (**28**) and catechin (**19**) as the flavonoids responsible for the yellow color of *D. graveolens* flowers. These two metabolites are present in aqueous residue in a quantity of 4 mg/g and 5.92 mg/g of extract, respectively. The high flavonoid content means that the aqueous residue can be reused for its good dyeing power in the dyeing process [11].

Previously, Lanzetta et al., in 1991 [14], identified two xanthanolides in an acetone extract of leaves, together with sesquiterpenes. Compounds were isolated on Sephadex LH-20 columns followed by fraction purification using TLC; thus, compounds were identified with EIMS and NMR methods.

In 2018, Silinsin et al. studied the composition of plant leaves’ ethanol and water extracts, identifying 10 phenolic compounds among 27 standards. The major phenolic compounds present in leaves were chlorogenic (**7**) and quinic (**14**) acids (2167 ± 106 and 845 ± 41 ppb, respectively) [32].

In 2007, the ethanol extract of *D. graveolens* flowers, after a TLC examination, showed to contain terpenoids, as well as coumarins, phenolics, and flavonoids; however, alkaloids were not identified [12].

Afterward, the crude CH_2_Cl_2_/MeOH extract of the air-dried epigeal parts of *D. graveolens* was subjected to fractionation and further separation and purification by Abou-Douh in 2008. Thanks to structure elucidation procedures, several known sesquiterpenes and two new eudesmane sesquiterpene derivatives, 3*a*-hydroxyilicic acid methyl ester (**171**) and 2*a*-hydroxy-4-*epi*-ilicic acid (**173**) [13], were found in addition to the ones previously identified in aerial parts by Sevil et al. in 1992 [17].

## 3. Biological Activities

The use of natural medicines has for centuries been the only way to treat human illnesses. Nowadays, plant-based phytochemicals are viewed as promising compounds for treating or preventing several diseases due to their safe characteristics. Genus *Inula* is widely used in East Asia for its several medicinal properties such as antibacterial, anticancerous, cytotoxic, hepatoprotective, and anti-inflammatory activity (Figure 3).

For this reason, researchers have focused their attention on the study of phytochemistry and the biological activity of this genus. In particular, preliminary investigations made on *D. graveolens* species have evidenced varied medicinal properties attributable to the followed active molecules: eugenol (**9**), aromadendrin, 7-O-methyl (**18**), quercetin (**28**), borneol (**40**), bornyl acetate (**41**), eucalyptol (**55**), *p*-mentha-1(7),2-dien-8-ol (*β*-phellandren-8-ol) (**76**), *α*-pinene (**86**), *α*-terpineol (**95**), carabrone (**122**), eudesma, 12-carboxy,3,11(13)-diene (**148**), ilicic acid (**169**), invalin (**177**), invalin acetate (**178**). This section summarized the latest available knowledge on the potential biological activity of *D. graveolens* (Table 1).

### 3.1. Antioxidant Activity

Overproduction of oxidants in the human body is responsible for oxidative stress, a pathological condition associated with several diseases such as diabetes, neurodegenerative, and cardiovascular ailments [34]. It has been reported that intake of vegetables and fruits, rich sources of bioactive molecules, could prevent or delay the development of chronic diseases due to their antioxidant properties [35].

As mentioned before, the harvesting period affected the phytochemical composition and therefore the biological activity of the plant extracts. Essential oils obtained from aerial parts of *D. graveolens* collected in the four seasons were evaluated by DPPH, ABTS, and *β*-carotene bleaching tests [30]. Essential oils from plants collected in April and July were the most active. Particularly, in summer, the essential oil showed the best radical scavenging activity vs. DPPH reporting an EC_50_ of 23.12 ± 0.29 µg/mL lower than the synthetic antioxidant butylhydroxytoluene (BHT) (31.2 ± 0.5 µg/mL), whereas in April, it demonstrated the strongest activity by ABTS (EC_50_ = 7.58 ± 0.4 µg/mL) and *β*-carotene bleaching assays (EC_50_ = 79.10 ± 0.59 µg/mL) [30].

The aqueous residue released during essential oil production was subjected to the radical-scavenging activity by DPPH and ORAC methods for an alternative application [11]. Results showed a significant radical scavenging activity vs. DPPH (IC_50_ = 0.022 mg/mL) comparable with that of quercetin (**28**) (IC_50_ = 0.013 mg/mL). Moreover, the extracts slowed the loss of fluorescence of fluorescein by quenching peroxyl radicals (ORAC) in a dose-dependent manner in the range 0.01 and 0.05 mg/mL, leading to AUC_net_ of 90 at 0.05 mg/mL.

*D. graveolens* was used to recover specialized compounds by different solvent and extraction procedures. Nowadays, different techniques are used for the extraction of the antioxidant compounds from plants: the traditional methods like maceration, Soxhlet extraction, and the innovative ones like microwave-assisted extraction (MAE) and ultrasound-assisted extraction (UAE) [36]. This latter allowed to improve the extraction yield of active compounds by reducing solvent, time, and energy. In the UAE, the mechanical effect of the ultrasound causes breaking the cell walls, so they release their content. MAE uses electromagnetic radiations that penetrate the vegetable matrix, leading to the rupture of the cells and the release of intracellular products into the solvent. The microwave effect induced an increase in the temperature and consequently a rapid completion of the reaction. Thus, extraction techniques influence the phytochemical composition and biological activity. As reported by Souri and Shakeri in 2020 [31], aerial parts of *D. graveolens* extracted with different percentages of methanol and water by microwave showed the highest yield of phenols and tannins. Consequently, it demonstrated the best antioxidant activity in DPPH assay, reporting an IC_50_ value of 7.7 mg/mL, lower than the extracts obtained by ultrasound-assisted extraction (UAE) (IC_50_ value 21.5 mg/mL) and maceration (IC_50_ value 32.3 mg/mL).

Methanolic extract of *D. graveolens* showed higher lipid peroxidation inhibition (64.28%, at 50 mg/mL), evaluated by *β*-carotene bleaching test [37], but it was lower than BHT used as standard, which at the same concentration reported 88.57% of antioxidant activity after 105 min at 50 °C [37]. Moreover, *D. graveolens* plant extract reported the best ability in chelating ferrous ions at 20 mg/mL (up to 96%). At the same concentration, the extract showed the best ability to scavenge the hydroxyl radicals generated by the Fenton reaction. The extract counteracted superoxide anion without significant difference among the doses reporting a scavenging percentage of 82.51% and 93.43% at 4 and 12 mg/L, respectively [37]. Leaf extract demonstrated better antioxidant activity than the whole plant extract, reporting interesting antioxidant effects at lower doses. As reported by Boudkhili et al. 2012 [38], *D. graveolens* leaf extract inhibited the oxidation of *β*-carotene, reporting 45% of antioxidant activity after 24 h at 2 mg/mL, a dose of about 10 times lower than that previously reported [37]. In addition, leaf water and ethanol extracts showed DPPH scavenging activity having an IC_50_ value of 29.1 µg/mL and 35.9 µg/mL, respectively [32].

### 3.2. Antitumor Activity

Many studies reported that natural products could represent new and alternative strategies for preventing and treating several human diseases characterized by excessive production of free radicals involved in several diseases, including cancer, and that it is important to have a holistic approach [39,40].

Essential oils of aerial parts of *D. graveolens* and the main compound, bornyl acetate (**41**), showed an IC_50_ of 66.5 and 85.6 µg/mL, respectively, lower than *cis*-platin (IC_50_ 141.5 µg/mL) used as standard. Essential oils demonstrated the highest toxicity on HT-29 (IC_50_ 24.6 µg/mL) and on A549 (IC_50_ 28.3 µg/mL) cell lines. In all cell lines, the essential oil was more active than bornyl acetate (**41**) and the standard *cis*-platin. The disadvantage was that both essential oils and bornyl acetate (**41**) also affected normal cells (human amnion cells, also indicated with the acronym of FL), reporting an IC_50_ of 42.1 and 50.6 µg/mL, respectively. Morphologically, both caused disruption and disintegration of cells and detachment of cells from the plate surface in a dose-dependent manner [24]. Otherwise, no toxicity was observed on fibroblasts [11,41]. Bornyl acetate, as other monoterpenes, demonstrated the cytotoxic effect by promoting apoptosis generally caused by a high amount of ROS, and cytostatic effect by inducing cell cycle arrest in the G2/M phase leading to the inhibition of cell invasion and migration [42]. The aqueous residue of the hydrodistillation of *D. graveolens* reported no cytotoxic effect on skin healthy human fibroblast CCD-45 SK at all tested doses after 72 h of treatment [11] as well as polycaprolactone (PCL) polymeric scaffold made with methanol extract of *D. graveolens* aerial parts on fibroblasts (rat dermal) cell line after 24, 48 and 72 h of treatment [41]. Moreover, the scaffold promoted cell proliferation compared to the control group and cells treated with a scaffold made of PCL alone, suggesting the potential biomedical application of *D. graveolens* in tissue engineering, for example, to replace damaged tissues, support cell growth, possible injury healing [41]. Other compounds isolated from *D. graveolens* demonstrated cytotoxic activity against murine lymphocytic leukemia cells (P-338), nasopharyngeal carcinoma cells (KB-3), and vinblastine resistant cells (KB-V1) [33]. Ivalin (**177**) is the major compound found in many *Dittrichia* species and, with its derivative ivalin acetate (**178**), demonstrated the best activity on all investigated cell lines. In particular, ivalin (**177**) was the most cytotoxic compound showing an ED_50_ value ranging from 0.14 to 1.8 µg/mL on investigated cell lines (P-338, KB-3, KB-V1 cell line) [33]. Previous studies reported the ability of ivalin in promoting apoptosis by inducing Bax protein and increasing membrane mitochondrial permeability with the consequent release of cytochrome c into the cytosol [43]. Cytotoxicity was also confirmed on breast adenocarcinoma cell line (MCF7) [12]. Ethanol extract of flowers reported an IC_50_ of 3.83 µg/mL after 72 h, probably due to the presence of flavonoids and phenolic acids. Chloroform extract demonstrated similar toxicity (6.80 ± 1.73 µg/mL), unlike the aqueous extract which reported lower MCF7 cytotoxicity with an IC_50_ value ten times higher [12].

### 3.3. Antimicrobial Activity

In recent years, antibiotic resistance is becoming an important problem all over the world, and for this reason, the search for new molecules plays a key role. Plant metabolites as polyphenols, alkaloids, and terpenoids could represent an interesting approach to afford this task since they can destroy bacterial cells or inhibit their growth [44]. Several natural compounds or plant extracts, including *D. graveolens*, have been widely studied to assess the antibacterial activity against different species of fungi and bacteria.

In 2016, Mitic et al. [28], reported that the *D. graveolens* essential oil (10 mg) was effective against two *Gram*-positive bacteria, *Staphylococcus aureus* and *Bacillus subtilis*, showing an inhibition zone diameter respectively of 33.0 and 22.0 mm compared to 15 μg of streptomycin (23 mm) and 30 μg of chloramphenicol (26–30 mm) used as references [28]. The antibacterial activity of *D. graveolens* on *S. aureus* was also investigated by Guinoiseau et al. [45], which reported a minimum inhibitory concentration (MIC) value of 5 mg/mL and a minimum bactericidal concentration (MBC) value of 10 mg/mL, very close to each other thus showing a bactericidal activity. In addition, the preliminary treatment with the essential oil at MIC concentration showed that after 2 h of exposure, the bactericidal end-point (99%) was obtained. They observed that the mechanism responsible for this action involved the uncommon invagination of the cell wall and the alterations in the cytoplasm density and distribution, which represent the potential sites of the antibacterial action of the substances [45]. Bamuamba et al. [46] also reported the antibacterial activity of the acetone/water (4:1) crude extract and its hexane fraction against *S. aureus*. In contrast, a recent study reported no activity against *S. aureus*, while *D. graveolens* extract (1.25 mg) was able to inhibit the *Gram*-positive *B. subtilis* and *Enterococcus faecalis* with an inhibition zone of 10 and 34 mm respectively [31]. The extraction processes influence the chemical profile, which may explain the difference in the antibacterial activity results obtained between the extracts or essential oils.

The inhibitory effect of the essential oil against different planktonic strains of *E. faecalis* was also reported by Benbelaid, et al. [47], obtaining a diameter inhibition zones ranging from 12 ± 1 and 13 ± 1 mm and MIC between 2.000 ± 0.000 and 4.000 ± 0.000% *v/v* against all tested strains [47]. However, no antibacterial activity was reported by Boudkhili et al. [38] against *B. subtilis*, *Escherichia coli*, *Micrococcus luteus*, *Salmonella* spp, *Staphylococcus* spp.

Another study showed that the *Gram*-positive *B. cereus* was inhibited by *D. graveolens* crude extract CH_2_Cl_2_/MeOH (1:1) and its five fractions (Et_2_O/hexane 1:3; Et_2_O/hexane 1:1; Et_2_O/hexane 3:1; Et_2_O; Me_2_CO/Et_2_O 1:1) at the concentration of 200 μg/mL with inhibition zone values ranged between 15 and 20 mm [13].

To date, there are only a few studies reporting the antibacterial activity of the *D. graveolens* essential oil against *Gram*-negative bacteria. Usually, *Gram*-negative bacteria show greater resistance to essential oils than *Gram*-positives because of the different structure of the cell walls; thus, the mechanism of action of the essential oils or their compounds depend on their chemical properties [48,49]. In particular, the hydrophobicity of substances is important for the activity as it guarantees the disruption of bacterial structures with a consequent increase in permeability [49].

The essential oil was tested against *Salmonella thypi* and *Enterobacter aerogenes*, causing an inhibition zone diameter of 33 and 27 mm, respectively [31]. However, it did not inhibit the *Gram*-negative bacteria *E. coli* and *P. aeruginosa,* contrary to what Bamuamba et al. have reported [46].

Djenane et al. [21], on the other hand, have reported the antibacterial activity against the *Gram*-negative *Campylobacter jejuni*, responsible for foodborne and gastrointestinal tract infections. Specifically, the essential oil showed an inhibition zone diameter of 53.3 ± 9.0 mm compared to 21 ± 2.6 mm of the positive control gentamicin with a MIC value of 0.2 ± 0.02% *v/v*. Furthermore, the authors investigated the *C. jejuni* growth inhibition in chicken meat during storage. In particular, the essential oil (at 2-fold MIC concentration) showed a reduction of 3.08 log_10_ cfu/g after four days of storage and 6.94 log_10_ cfu/g after eight days compared to untreated samples (initial populations 5.60 log_10_ cfu/g increased to 8.14 log_10_ cfu/g at the end of the storage) thus contributing to the preservation of the chicken meats [21].

The *D. graveolens* aerial parts essential oil was also tested for antimycotic activity. In particular, it was effective against ten isolates of *Candida albicans* with a total MIC of 30.675 mg/mL. The mechanism of action involves the lipophilicity and volatile nature of essential oil compounds, which help to attach and penetrate cell membranes [10]. Abou-Douh et al. [13], also investigated the antifungal activity showing that the crude extract CH_2_Cl_2_/MeOH (1:1), and its two fractions at 200 μg/mL (Et_2_O/hexane 1:3 and Me_2_CO/Et_2_O 1:3), possess the same activity as the positive control tioconazole (10 mm) against the fungus *Scopulariopsis brevicaulis*, while other two fractions (Et_2_O and Me_2_CO/Et_2_O 1.5:8.5) exerted a higher effect than tioconazole (14 and 12 mm respectively vs. 10 mm of the positive control) [13].

The presence of compound classes such as alcohols, aldehydes, and phenolics is associated with antimicrobial activity [50]. In addition to these compounds, sesquiterpene acids, and lactones are also reported as antimicrobials. Topҫu et al. [33] investigated the activity of the flavonoid 7-*O*-methylaromadendrin (**18**), and the two sesquiterpenes inuviscolide (**204**) and carabrone (**122**) extracted from *D. graveolens* aerial parts. They reported weak antibacterial activity against *S. epidermidis* with MIC values of 40, 100, and 80 μg/mL respectively if compared with streptomycin (MIC 1.6 μg/mL) and penicillin G (MIC ≤ 0.02 μg/mL). Furthermore, ivalin (**177**), a sesquiterpene lactone was active against *B. subtilis* (MIC 375 μg/mL) but less than the positive controls amoxicillin (MIC 0.25 μg/mL) or gentamicin (MIC ≤ 4 μg/mL) [33]. The terpenes borneol (**40**) and bornyl acetate (**41**), the main constituents of *D. graveolens* essential oil, have also been reported weak antibacterial activity [28,50].

However, the chemical profile and thus the biological activities of plant extracts depend on seasonal changes. Sellem et al. [30] demonstrated that the antibacterial effect of *D. graveolens* essential oil produced in the autumn was greater against the tested microorganisms. In particular, the treatment with 1 mg/mL of the autumn essential oil has reported the lowest MIC value (15.6 μg/mL) against *S. aureus* and *M. luteus*, while the July oil was the most effective against the fungal pathogen *C. albicans* (MIC = 250 μg/mL) [30].

It is difficult to attribute the effect to a single compound; the activity is usually determined by a synergy between the components of the essential oil blend [51]. In addition, in order to improve the antimicrobial effect and to reduce antibiotic doses, essential oils could be associated with antibiotics. Miladinović et al. [26] reported that the combination of *D. graveolens* essential oil with chloramphenicol, showed a 10-fold reduction of the antibiotic MIC (1.0 to 2048.0 μg/mL) against different tested bacteria may be due to the borneol (**40**) and other constituents of the essential oil that favor the entrance of the antibiotic. Instead, the association with tetracycline decreased the MIC (0.5 to 64.0 μg/mL) from 1.7-fold to 3.3-fold. In this way, it is possible not only to enhance the antimicrobial activity of the essential oil but also to reduce the dose and the adverse side effects of antibiotics, thus representing a strategy to overcome antibiotic resistance [26].

The antibacterial activity was also investigated in the aqueous residue of the hydrodistillation process to obtain the essential oil. Gharred et al. [11] reported that the extract (10 mg/mL) was active against *Vibrio parahaemolyticus*, *Vibrio alginolyticus,* and *S. epidermidis,* producing an inhibition zone diameter of 18, 21, and 17 mm respectively after 18h of incubation. This represents a great alternative since the aqueous residue is a source of bioactive compounds that can be exploited for biological activities avoiding the risk of environmental pollution [11].

### 3.4. Anti-Inflammatory Activity

In addition to the activities previously reported for the aqueous residue of *D. graveolens* obtained by hydrodistillation, significant anti-inflammatory activity was also demonstrated [11]. In fact, in a recent in vivo study, the aqueous extract (5 and 10 mg/kg), or reference drug (dexamethasone and Aspegic^®^, 15 mg/kg) were intraperitoneally administrated in mice model and, 30 min after, the xylene, as phlogogenic agent, was topically applied to the right ear of mice. The Aspegic^®^ reference showed the highest anti-inflammatory potential (91.48%) [11]. By comparison with the control, aqueous extract suppressed the ear edema in a dose-dependent manner, reaching 75.59% of inhibition (10 mg/kg), higher than dexamethasone (53.49%). The anti-inflammatory activity is linked to the presence of terpenes which could influence the activities of crucial mediators of inflammation. In fact, it is reported that borneol attenuates the activity of iNOS and COX-2 whose inhibition reduces the production of arachidonic acid, leukotrienes, and prostaglandins [52]. In addition, also the sesquiterpenes inuviscolide (**204**) and ilicic acid (**169**) have been reported for their interference in leukotrienes synthesis [53].

### 3.5. Anti-Cholinesterase and Anti-Tyrosinase Activity

Essential oil of *D. graveolens* was found to be able to inhibit acetylcholinesterase (AChE) and tyrosinase, enzymes involved in the neurodegenerative and depigmentation disorders, respectively, in humans [54].

AChE is predominant in the brain of healthy individuals regulating the acetylcholine level. This enzyme, along with butyrylcholinesterase, is a potential target to ameliorate the cholinergic lack in Alzheimer’s disease. Nowadays, the research of inhibitors of AChE from natural matrices has gained more attention. The acetylcholinesterase inhibition of *D. graveolens* essential oil (from France) was reported for the first time by Dohi et al. [22] with an IC_50_ value of 0.27 ± 0.10 mg/mL by in vitro microplate assay method. Recently, more interesting results reported an increased inhibitory capacity of essential oil from Tunisian plants collected from April (IC_50_ 5.50 ± 0.25 µg/mL) to October (IC_50_ 5.01 ± 0.34 µg/mL), then decreased in Winter (IC_50_ 8.12 ± 0.54 µg/mL) [30]. The activity could be affected by chemical composition related to the harvesting season and the plant origin.

In Autumn, the content of α-terpineol (**95**) (2.74%, IC_50_ 1.3 ± 0.06 mg/mL for AChE inhibition) was higher than other seasons (0.55–1.71%) and in comparison, with French cultivar (1.6%). The presence of other molecules with anti-AChE activity, as 1,8 cineol (**55**) (IC_50_ 0.015 ± 0.003 mg/mL), eugenol (**9**) (IC_50_ 0.48 ± 0.16 mg/mL) and *α*-pinene (**86**) (IC_50_ 0.022 ± 0.003 mg/mL) could contribute to the biological activity [16,22,30].

Other Inula species reported anti-AChE activity; results ranged from 3.56 ± 0.16 mg GALAE/g) (*I.* *peacockiana)* to 5.13 ± 0.15 mg GALAE/g) (*I. aucheriana).* Some sesquiterpene lactones from *I. oculus-christi* and *I. aucheriana* (gaillardin, pulchellin C, and britannin), not identified in *D. graveolens*, inhibited in vitro AChE enzyme, but the mechanism was not investigated [55].

Tyrosinase plays a vital role in the enzymatic browning of food and depigmentation disorders in humans, playing a key role in the synthesis of melanin [54]. *D. graveolens* also showed a weak in vitro tyrosinase inhibitory activity in comparison with kojic acid as a reference standard (IC_50_ 4.05 ± 0.25 µg/mL). Harvesting season also affected this inhibitory activity, with IC_50_ values ranging from 18.34 ± 0.21 µg/mL (April) to 49.25 ± 0.5 µg/mL (July) [30]. Previous studies reported that the sesquiterpenes as 1-*O*-acetylbritannilactone [56] and inulavosin [57] from *Inula britannica* L. and *Inula nervosa* Wall., respectively, were found to be tyrosinase inhibitors in cell-based systems. The compound 1-*O*-acetylbritannilactone from *I. britannica* inhibited cell-based tyrosinase activity in a dose-dependent manner in B16 melanoma cells, but no evidence was reported on mushroom and mouse tyrosinase activity by cell-free assay. Thus, its potential mechanism is not directly related to the catalytic activity of the enzyme [56]. Other *Inula* species, *I. peacockiana* and *I. viscidula* methanol extracts inhibited the activity of tyrosinase, with 120.65 ± 0.35 mg KAE/g and 122.13 ± 0.63 mg KAE/g [58].

The table below summarizes the biological activities of *D. graveolens* (Table 2).

## 4. Toxicology

Ethnobotanical studies reported the use of *D. graveolens* for treating cold and against bruises and burns, as external use. Further, the anti-hemorrhoidal employment of this drug and the possibility of using the leaf and twig infusion to treat diabetes and high blood pressure were documented [59,60]. The employment of this species as a traditional medicinal plant was confirmed by investigating its biological activity according to the harvested season, and it was demonstrated that the most interesting biological activity was seen for *D. graveolens* species collected in April and July [30].

However, besides the health benefits, irritant and/or allergic dermatitis cases are also reported from *D. graveolens.* Iconic for this adverse reaction is a case study of a 56-year-old man showing dermatitis with marked erythema and hyperkeratosis on the dorsum of the hands and, to a lesser extent, hyperkeratosis on the knees and elbows. The patch test performed by the North American Contact Dermatitis Research Group recognized plants from the Compositae family and, in particular, *D. graveolens* as triggering factors for dermatitis. The allergen was primarily identified in inuviscolide (**204**), a sesquiterpene lactone. Molecules from this chemical class are common in many plants belonging to the Compositae family; this can also explain the positive patch test not only for *D. graveolens* but also for *Frullania*, *Laurus nobillis*, and other Compositae. The patient’s skin lesions were treated with clobetasol cream, a class I topical corticoid, and vinyl gloves occlusion at bedtime [61].

Other adverse effects related to *D. graveolens* were observed in sheep. Specifically, after *D. graveolens* seed ingestion, pyogranulomatous enteritis and enterotoxemia were observed due to bristles penetration into the intestine mucosa. This enteritis may contribute to cases of undernutrition, anemia, illthrift, and deaths in flocks grazing in fields where this weed grows. Philbey and Morton have observed the presence of penetrating bristles in the jejunum [62], while Schneider and Plessis have noted reddening and thickening across the small intestine of affected sheep [63]. The location of the penetrating bristles into the small intestine can be explained by the fact that they may pass intact throughout the forestomaches and abomasum before separating from the seed for partial digestion in the small intestine. This process allows bristles to embed in the mucosa. However, it was seen that after six days from *D. graveolens* removal as food, clinical signs of enteritis disappeared even if bristles were still in sheep intestinal mucosa [62]. In another study, Seddon and Carne failed to induce diseases in sheep by feeding them with *D. graveolens*; in this case, it was not used mature seed but vegetative growth and finely ground pappus hair [64]. In general, it is possible to say that pastures containing *D. graveolens* should be grazed in the vegetative stage, while when it is in the seeding stage, the presence of an alternative feed is necessary.

The toxic effect of *D. graveolens* was also demonstrated in fishes. The aerial parts of this species are occasionally used to facilitate freshwater fish capture by fishermen. In fact, after maceration in water, the immersion of leaves in the fishing site led to definitive sedative effects of fishes’ present in the surrounding area. This ichthyotoxicity seems to be related to the presence in *D. graveolens* of two major sesquiterpenes 12-carboxyeudesma-3,11(13)-diene (**148**) and tomentosin (**189**) [14]. Leaves essential oil from *D. graveolens* was also investigated, together with other 18 essential oils, to evaluate its repellent capacity against *Aedes aegypti* mosquitoes. It was found that although to a lesser extent than more potent essential oils, *D. graveolens* showed a reduction in mosquitoes’ attraction to human fingers when compared to the finger used as a unique stimulus. This repellent activity seems to be related to the presence of bornyl acetate (**41**) and *p*-mentha-1(7),2-dien-8-ol (**76**) in *D. graveolens* essential oil [65].

Finally, the allelopathic activity of *D. graveolens* was also investigated. Omezzine et al. demonstrated that the incorporation of *D. graveolens* shoots and flowers powder in the soil culture of *Lactuca sativa* L., *Raphanus sativus* L., *Peganum harmala* L., and *Silybum marianum* L. significantly decreased the shoot and root length of this target species. Equally, the application of *D. graveolens* shoots and flowers aqueous extract in soil reduced the seedlings’ length even if the flower extract possesses a higher inhibitory effect than the shoot extract [66]. These results agree with that obtained by Abu Irmaileh et al., who demonstrated that the ethanolic extract of *D. graveolens* significantly reduced root length more than shoot length or seed germination at 200 ppm [67]. This is in line with knowledge for which root growth sensitivity is the best indicator of allelochemicals phytotoxicity for its high permeability to these chemicals [66,68]. The bio-guided fractionation of *D. graveolens* ethanolic extract allowed the isolation of allelopathic sesquiterpene ilicic acid (**169**). It was, indeed, demonstrated that this compound inhibited the growth of some plants at 25 ppm. In particular, ilicic acid (**169**) reduced the root length of cauliflower, cress, and radish [67]. Data obtained by these studies are in line with the invasive nature of *D. graveolens*. In fact, recently a theory has emerged for which the invasive nature of plants should be linked to their ability to produce secondary metabolites able to inhibit the growth of other species leading to the elimination of competitive vegetation [69,70].

## 5. Materials and Methods

This study aims to review all the metabolites and biological activities of *D. graveolens* reported in the literature until October 2021. This review included articles found on two specific databases: Scopus and PubMed. Only English articles containing the selected keywords (*Inula graveolens* and *Dittrichia graveolens*) were detected, while the unavailable full texts were not requested. The initial selection provided 85 articles, of which 68 were found in Scopus and 17 on PubMed. Among the 85 articles, 52 were relevant to the research topic. All selected articles were carefully analyzed and divided by argument (chemical characterization, biological activity, toxicology), as shown in Figure 4.

## 6. Conclusions

*D. graveolens* is a Mediterranean native plant, spreading northward in Europe. It has recently attracted increasing research interest due to its chemical composition and biological activities, especially due to its terpene and phenolic content. Moreover, it has been demonstrated that the chemical composition is strongly influenced by plant collection seasons, phenological stage, processing, and extraction methods. Among studied biological activities: antioxidant, antimicrobial, cytotoxic, and cholinesterase inhibitory activities seem to be the most promising. Although this plant species has been used for ages as traditional medicine for different purposes, several areas still need investigation like antirheumatic, anti-inflammation, and anti-infection against Leishmaniasis. In fact, more studies are needed, as preclinical or clinical studies, to define the molecular pathways involved in biological activities demonstrated by *D. graveolens* as well as the role of the main components for its pharmacological application. Moreover, it is necessary to determine whether there might be synergy or antagonism among these components and, at the same time, there might be other industrial applications of its extract or essential oil in the food industry, for its effectiveness as an antioxidant or antimicrobial agent.

Furthermore, the ecological role of *D. graveolens* seems to be worth future investigation as the invasive nature of plants should be linked to their ability to produce specialized metabolites able to inhibit the growth of other species leading to the elimination of competitive vegetation and to the spread of climate change. Moreover, characterization of plant biology and life cycle traits, including growth and phenology, is a necessary first step for assessing invasion potential and for developing targeted management strategies for invasive plants.

Despite all the studies carried out, there are still several open issues, and there is a lack of description of its bioavailability that has not been thoroughly investigated. Consequently, further studies are needed to address the ecological role and mechanism of action of *Dittrichia*.

## Figures and Tables

**Figure 1 molecules-27-00895-f001:**
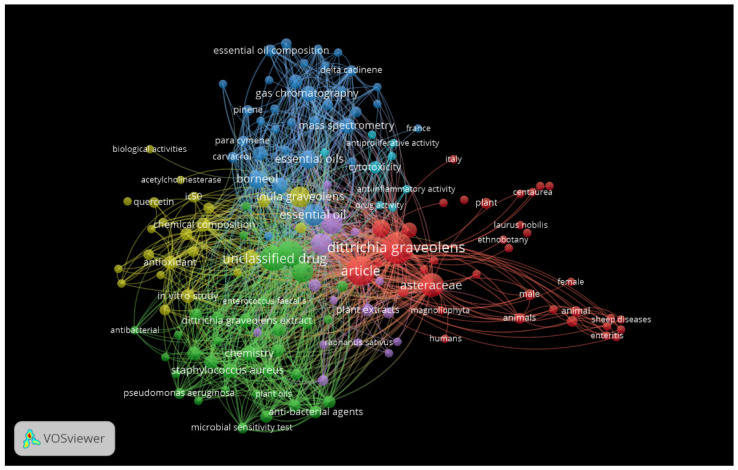
Bubble map visualizing items from articles included in the review.

**Figure 2 molecules-27-00895-f002:**
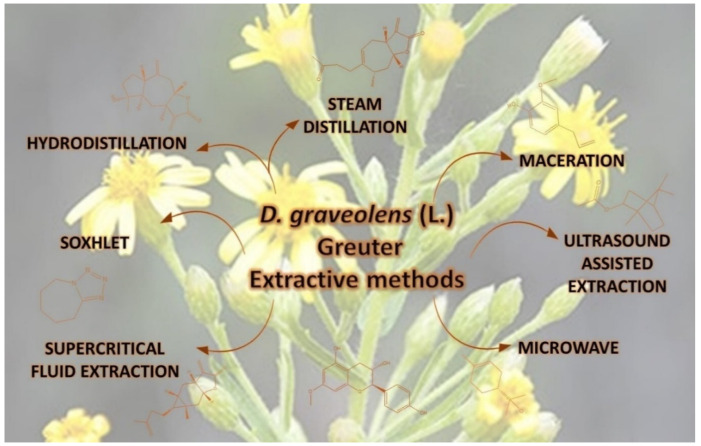
Methods used for the extraction of active principles from *D. graveolens*.

**Figure 3 molecules-27-00895-f003:**
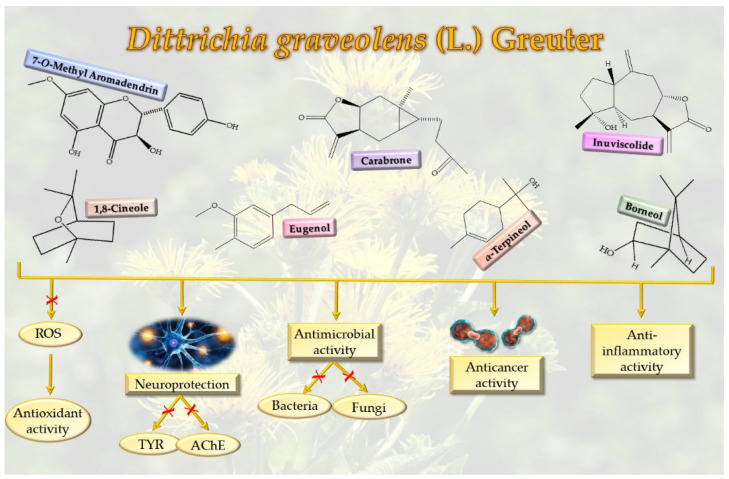
Graphical representation of *D. graveolens* biological activities. Red marks indicate the blocking of ROS formation, inhibition of tyrosinase (TYR) and acetylcholinesterase (AChE), and protection against bacteria and fungi by *D. graveolens*.

**Figure 4 molecules-27-00895-f004:**
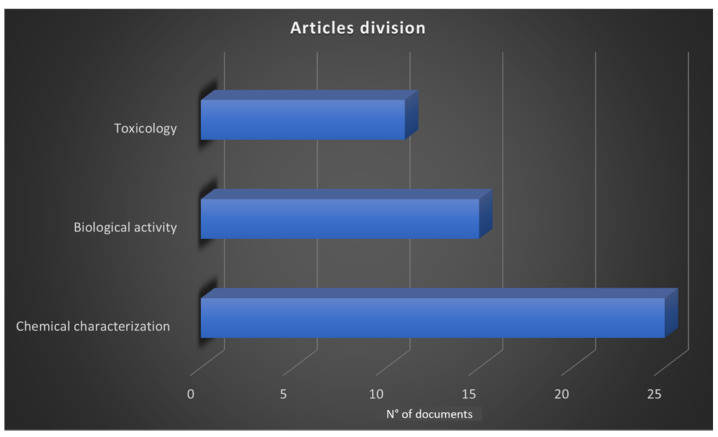
Selected articles divided per argument.

**Table 1 molecules-27-00895-t001:** Bioactive metabolites present in *D. graveolens* L.

Bioactive Compounds								
Compound	Formula	Structure	Analyzed Sample	Extraction Method	Chemical Composition Analysis	Quantitative	Origin of Plant	Reference
Eugenol (**9**)	C_10_H_12_O_2_	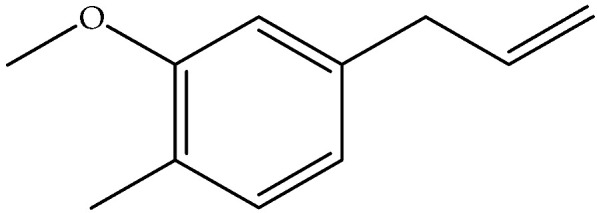	Essential oil	Steam-distilled	GC-MS	Trace	France	[16,22]
Aromadendrin, 7-*O*-methyl (**18**)	C_16_H_14_O_6_	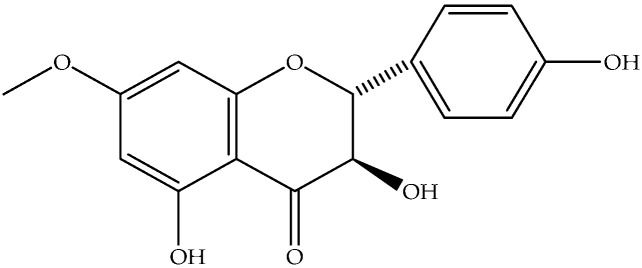	Air-dried and powdered aerial parts	Maceration with petrol- Et_2_O-MeOH (1:1:1), then with MeOH	Sephadex LH-20 columns, prep. TLC, EIMS, NMR	470 mg/32 g of the residue of extract treated with MeOH/1 Kg of initial aerial parts	Aydos Dag (Istanbul)	[17]
Quercetin (**28**)	C_15_H_10_O_7_	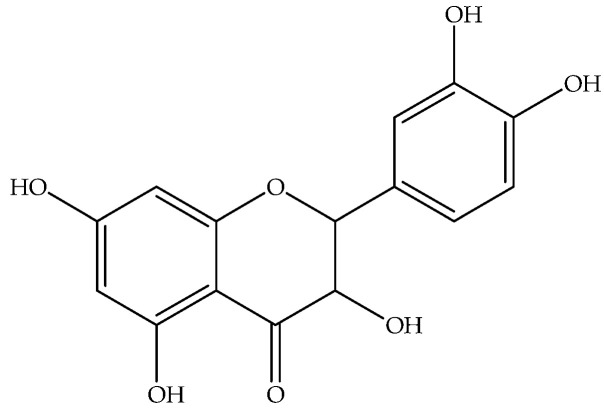	Aqueous residue of dried leaves and flowers hydrodistillation	Hydrodistillation	HPLC UV-vis	4 mg/g of extract	Monastir (Tunisia)	[11]
			Dried leaves	Maceration with with MeOH	UHPLC-MS/MS	118 ± 8 μg/kg extract	Bingol (Turkey)	[32]
Borneol (**40**)	C_10_H_18_O	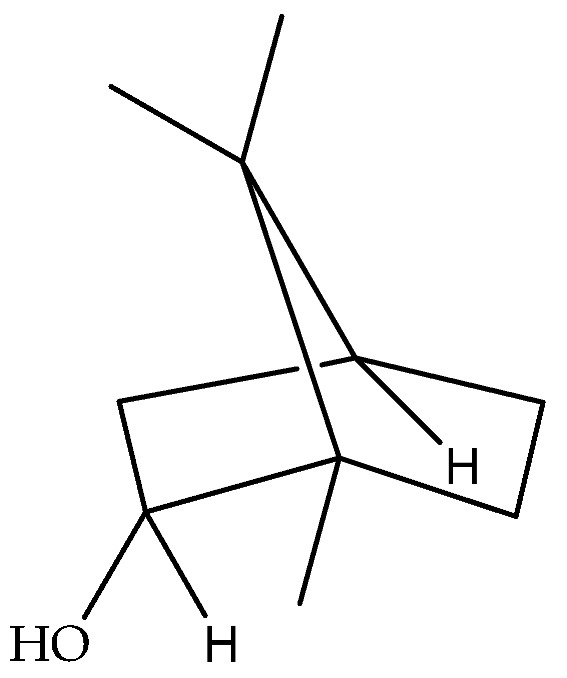	Essential oil	Hydrodistillation of air-dried aerial parts	GC-MS	5.44%	Shush (Khuzestan Province, Iran)	[10]
				Vapour distillation of aerial parts	GC-MS	7.6%	Ajaccio (Corsica)	[19]
				Hydrodistillation of fresh aerial parts	GC-MS	2.74–12.40%	Bekaa and Sannine (Lebanon)	[23]
				Hydrodistillation of dried aerial parts	GC-MS	11.34%	Tokat Turkey Provence	[24]
				Steam distillation of air-dried aerial parts	GC-MS	60.7%	Gorgan province (Iran)	[27]
				Hydrodistillation of air-dried aerial parts	GC-MS	12.8%	Attiki County (Greece)	[29]
				Steam distillation of fresh plant	GC-MS	23.65–37.29%	Chebba (Mahdia, Tunisia)	[30]
				Steam hydrodistillation of dried leaves	GC-MS	21.04%	Tizi-Ouzou province (Algeria)	[21]
				Steam-distilled	GC-MS	20 ± 0.35%	France	[22]
				Hydrodistillation of dried aerial parts	GC-MS	18.7%	Stara planina mountain (Serbia)	[26]
				Hydrodistillation of dried flowering aerial parts	GC-MS and GC-FID	43.6%	Village Vladimirovci (Montenegro)	[28]
				Hydrodistillation of the fresh aerial parts	GC-MS	18.3%	Constantine (North Easter Algerian)	[20]
Bornyl acetate (**41**)	C_12_H_20_O_2_	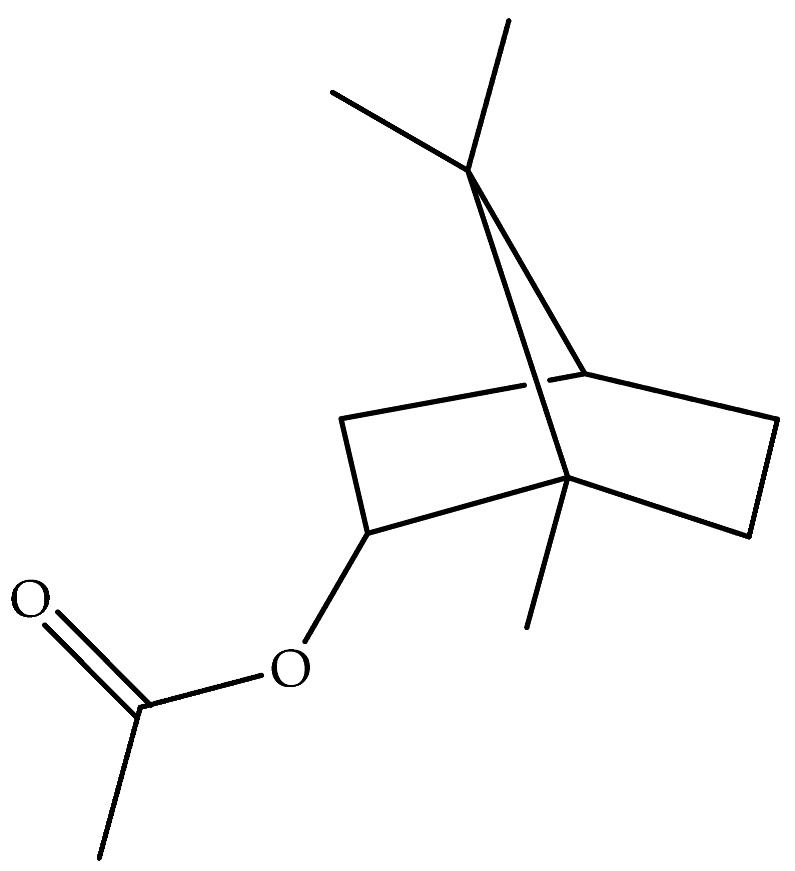	Essential oil	Hydrodistillation of air-dried aerial parts	GC-MS	0.24%	Shush (Khuzestan Province, Iran)	[10]
				Vapour distillation of aerial parts	GC-MS	56.8%	Ajaccio (Corsica)	[19]
				Hydrodistillation of fresh aerial parts	GC-MS	70.64–72.31%	Bekaa and Sannine (Lebanon)	[23]
				Hydrodistillation of dried aerial parts	GC-MS	60.43%	Tokat Turkey Provence	[24]
				Steam distillation of air-dried aerial parts	GC-MS	6.8%	Gorgan province (Iran)	[27]
				Hydrodistillation of air-dried aerial parts	GC-MS	25.4%	Attiki County (Greece)	[29]
				Steam distillation of fresh plant	GC-MS	40.16–45.34%	Chebba (Mahdia, Tunisia)	[30]
				Steam hydrodistillation of dried leaves	GC-MS	40.85%	Tizi-Ouzou province (Algeria)	[21]
				Steam-distilled	GC-MS	54 ± 2.0%	France	[22]
				Hydrodistillation of dried aerial parts	GC-MS	21.7%	Stara planina mountain (Serbia)	[26]
				Hydrodistillation of dried flowering aerial parts	GC-MS and GC-FID	38.3%	Village Vladimirovci (Montenegro)	[28]
				Distillation	NMR	57.4%	-	[18]
Cineole, 1,8 (syn. Eucalyptol) (**55**)	C_10_H_18_O	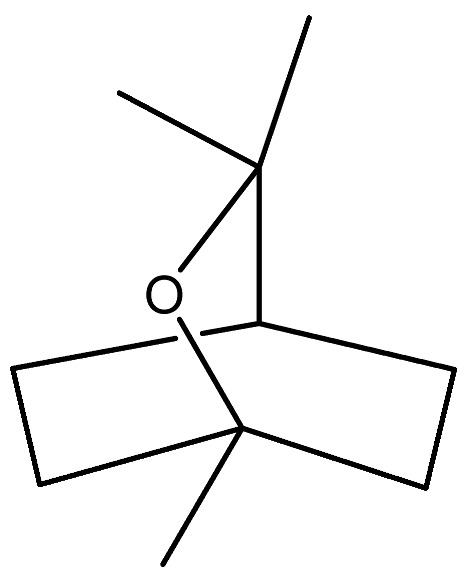	Essential oil	Hydrodistillation of air-dried aerial parts	GC-MS	54.89%	Shush (Khuzestan Province, Iran)	[10]
				Steam distillation of fresh plant	GC-MS	Trace—0.22%	Chebba (Mahdia, Tunisia)	[30]
				Steam hydrodistillation of dried leaves	GC-MS	2.41%	Tizi-Ouzou province (Algeria)	[21]
				Hydrodistillation of the fresh aerial parts	GC-MS	0.9%	Constantine (North Easter Algerian)	[20]
				Steam-distilled	GC-MS	Trace	France	[16,22]
			Supercritical fluid extract	Supercritical fluid extraction (SFE) of air-dried aerial parts	GC-MS	Trace	Monserrato (southern Sardinia)	[25]
*p*-Mentha-1(7),2-dien-8-ol (*β*-phellandren-8-ol) (**76**)	C_10_H_16_O	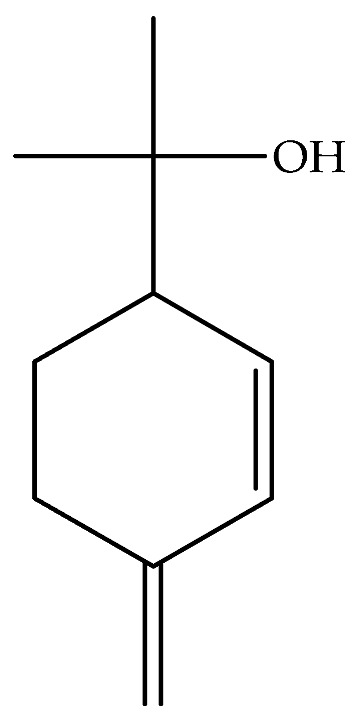	Essential oil	Vapour distillation of aerial parts	GC-MS	0.3%	Ajaccio (Corsica)	[19]
*α*-Pinene (**86**)	C_10_H_16_	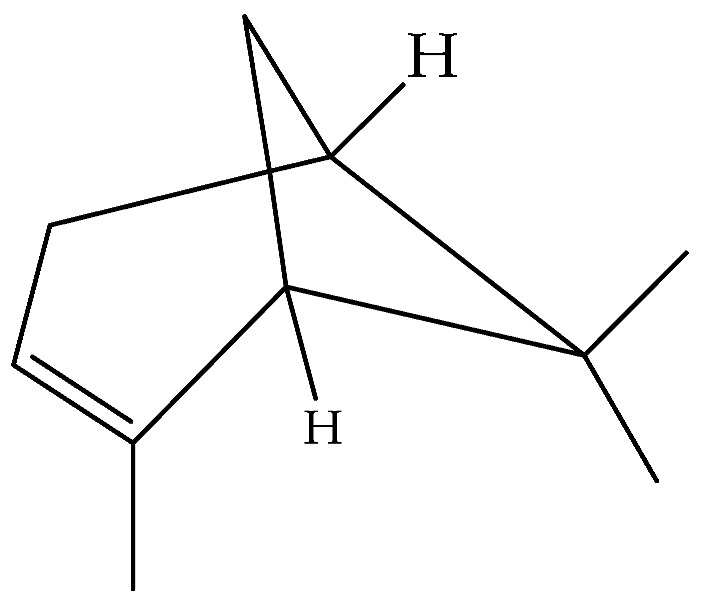	Essential oil	Hydrodistillation of air-dried aerial parts	GC-MS	3.21%	Shush (Khuzestan Province, Iran)	[10]
				Vapour distillation of aerial parts	GC-MS	0.3%	Ajaccio (Corsica)	[19]
				Hydrodistillation of fresh aerial parts	GC-MS	0.02–0.03%	Bekaa and Sannine (Lebanon)	[23]
				Hydrodistillation of dried aerial parts	GC-MS	0.16%	Tokat Turkey Provence	[24]
				Steam distillation of air-dried aerial parts	GC-MS	0.2%	Gorgan province (Iran)	[27]
				Hydrodistillation of air-dried aerial parts	GC-MS	Trace	Attiki County (Greece)	[29]
				Steam distillation of fresh plant	GC-MS	Trace–0.85%	Chebba (Mahdia, Tunisia)	[30]
				Steam-distilled	GC-MS	0.21 ± 0.0050%	France	[22]
				Hydrodistillation of dried aerial parts	GC-MS	1.2%	Stara planina mountain (Serbia)	[26]
				Hydrodistillation of dried flowering aerial parts	GC-MS and GC-FID	Trace	Village Vladimirovci (Montenegro)	[28]
*α*-Terpineol (**95**)	C_10_H_18_O	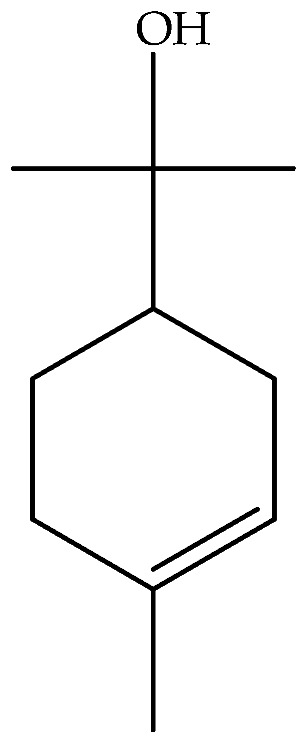	Essential oil	Hydrodistillation of air-dried aerial parts	GC-MS	1.31%	Shush (Khuzestan Province, Iran)	[10]
				Vapour distillation of aerial parts	GC-MS	0.4%	Ajaccio (Corsica)	[19]
				Steam distillation of fresh plant	GC-MS	Trace–0.85%	Chebba (Mahdia, Tunisia)	[30]
				Steam hydrodistillation of dried leaves	GC-MS	1.52%	Tizi-Ouzou province (Algeria)	[21]
				Steam-distilled	GC-MS	1.6 ± 0.062%	France	[16,22]
				Hydrodistillation of dried flowering aerial parts	GC-MS and GC-FID	Trace	Village Vladimirovci (Montenegro)	[28]
Carabrone (**122**)	C_15_H_20_O_3_	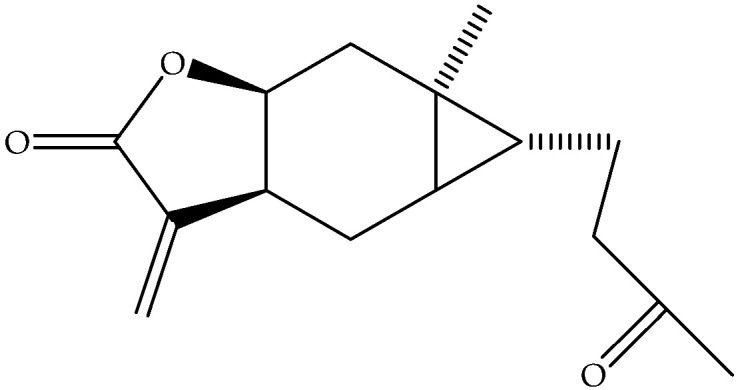	Air-dried and powdered aerial parts	Maceration with petrol- Et_2_O-MeOH (1:1:1), then with MeOH	Sephadex LH-20 columns, prep. TLC, EIMS, NMR	18 mg/32 g of residue of extract treated with MeOH/1 Kg of initial aerial parts	Aydos Dag (Istanbul)	[17]
Eudesma, 12-carboxy,3,11(13)-diene (**148**)	C_15_H_22_O_2_	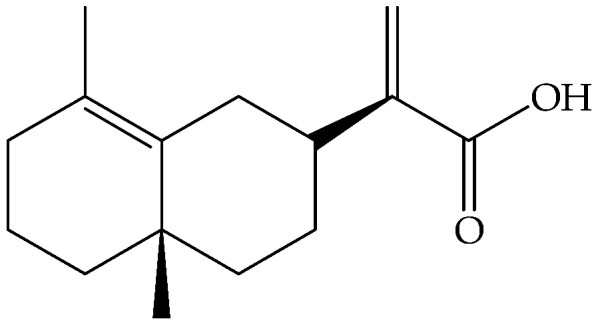	Fresh leaves	Maceration with Me_2_CO, then partition with different solvents	LC-MS and NMR	340 mg/1.3 g of petrol extract	Mediterranean area	[14]
Ilicic acid (**169**)	C_15_H_24_O_3_	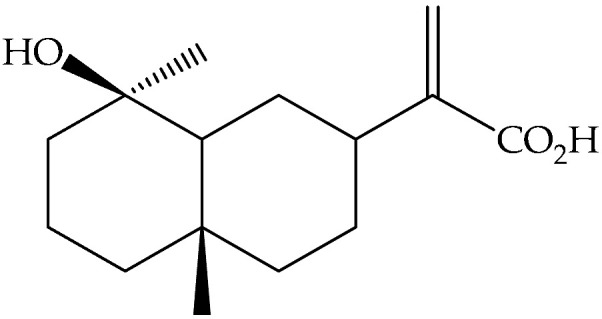	Air-dried and powdered aerial parts	Maceration with petrol- Et_2_O-MeOH (1:1:1), then with MeOH	Sephadex LH-20 columns, prep. TLC, EIMS, NMR	4800 mg/32 g of the residue of extract treated with MeOH/1 Kg of initial aerial parts	Aydos Dag (Istanbul)	[17]
			Air-dried epigeal parts extracts	Exhaustive maceration with CH_2_Cl_2_/MeOH followed by 80% MeOH	NMR	0.33%	Coastal regions of north-western Egypt	[13]
			Fresh leaves	Maceration with Me_2_CO, then partition with different solvents	LC-MS and NMR		Mediterranean area	[14]
Ivalin (**177**)	C_15_H_20_O_3_	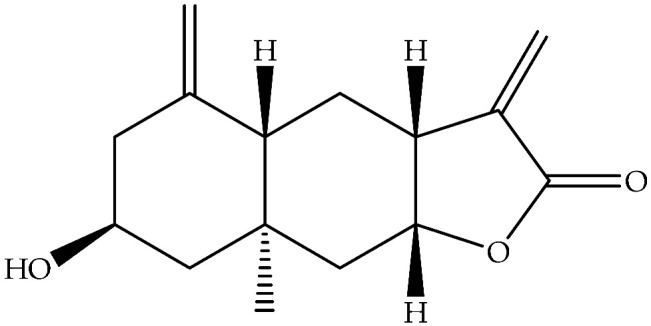	Air-dried aerial parts	Maceration with petrol-Et_2_O-MeOH	NMR		Aydos Mountain (Istanbul)	[33]
Ivalin, acetate (syn. Acetylivalin) (**178**)	C_17_H_22_O_4_	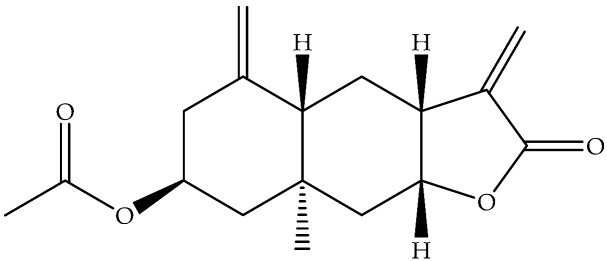	Aerial parts	Maceration with petrol-Et_2_O-MeOH	NMR		Aydos Mountain (Istanbul)	[33]

Electron Ionization Mass Spectroscopy (EIMS).

**Table 2 molecules-27-00895-t002:** Biological activity of *D. graveolens*.

Sample	Test/Model	Concentration/Dosage Tested	Effect	Reference
Essential oil from aerial parts	DPPH ABTS BCB	500; 250; 125; 60.25; 30.125 µg/mL	Antioxidant activity	[30]
	*S. aureus* *M. luteus* *C. albicans*		Antimicrobial activity	[10,30,45]
	*C. jejuni*	32 μL/mL to 0.3125 μL/mL	Antimicrobial activity	[21]
	Enzymatic assay	1 mg/mL to 0.0048 mg/mL	Acetylcholinesterase inhibition Tyrosinase inhibition	[22,30]
	HeLa, HT29, A549, MCF-7 cancer cells	0 μg/mL to 200 μg/mL	Antitumoral activity	[24]
Essential oil from leaves and flowers	DPPH ORAC	0 mg/mL to 0.12 mg/mL 0.01 mg/mL to 0.06 mg/mL	Antioxidant activity	[11]
	*V. alginolyticus*, *V. parahaemolyticus S. epidermidis*	10 mg/mL	Antibacterial activity	[11,47]
	*E. faecalis*	10 μL		
	Swiss mice	5 mg/kg and 10 mg/kg	Anti-inflammatory activity	[11]
Flowers essential oil	*S. aureus* *B.s subtilis*	10 mg	Antibacterial activity	[28]
Aerial parts extract	DPPH		Antioxidant activity	[31]
	*B. subtilis* *E. faecalis* *S. typhi* *E. aerogenes*	1.25 mg	Antibacterial activity	[31]
	*B. cereus* *S. brevicaulis*	200 μg/mL	Antibacterial activity	[13]
	Fibroblasts’ (rat dermal) cell line	5 wt.% of extract inserted on a scaffold made with polycaprolactone (PCL)	Cell proliferation promotion	[41]
	Male albino rats	1 mL/100 g body weight	Anti-inflammatory and antipyretic activity	[13]
Whole plant extract	BCB Reducing power Ferrous ion chelating ability Superoxide radical scavenging activity Hydroxyl radical scavenging activity	0 mg/mL to 20 mg/mL	Antioxidant activity	[37,38]
	*B. subtilis*, *Escherichia coli*, *Micrococcus luteus*, *Salmonella* spp., *Staphylococcus* spp.	10 mg/mL	No Antimicrobial activity	[38]
	*S. aureus*	62.5 μg/mL; 125 μg/mL; 250 μg/mL; 500 μg/mL; 1 mg/mL; 2 mg/mL	Antibacterial activity	[46]
Leaf extract	DPPH	30 μg/mL	Antioxidant activity	[32]
Flowers extract	MCF7 cell line	0.1 μg/mL to 100 μg/mL	Antiproliferative activity	[12]
Isolated compounds	P-338, KB-3, KB-V1 cell line	5 different concentrations not specified	Cytotoxic activity	[33]
	*S. epidermidis* *B. subtilis*		Antimicrobial activity	

Abbreviation: DPPH: 2,2-diphenyl-1-picrylhydrazyl; ABTS: 2,2’-azino-bis(3-ethylbenzothiazoline-6-sulfonic acid; ORAC: quenching peroxyl radicals; BCB: *β*-carotene bleaching tests; HeLa: human cervix carcinoma; HT29: human colon carcinoma; A549: human lung carcinoma; MCF7: human breast adenocarcinoma; P-388: murine lymphocytic leukemia; KB-3: nasopharyngeal carcinoma; KB-Vl: vinblastine resistant.

## Data Availability

Not applicable.

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
