# Peer review of "Dittrichia graveolens (L.) Greuter, a Rapidly Spreading Invasive Plant: Chemistry and Bioactivity"

_molecules, 2022, doi:10.3390/molecules27030895_

Round 1

Reviewer 1 Report

First of all, I would like to congratulate the authors of this review, as I think they have done a great job.

The importance of this review article on the chemistry and bioactivity of Dittrichia graveolens (L.) Greuter is duly justified and represents a valuable contribution to your field of knowledge. The information presented is well supported by reliable and meaningful references. All the characteristics presented on D. graveolens (L.) Greuter are based on verified scientific evidence.

The content of the review is exposed in a clear way. The data are presented in a quantitative, exact and contrasted way. The presentation of the content can be dense at times, since a large amount of information is included in the same section, as in section 2 “Phytochemistry”. I suggest adding subsections that promote content organization and facilitate understanding of the text. Also, the text of the figures can be difficult to read, I suggest enlarging the font of the typeface and / or increasing its sharpness.

The biological activities described in the review are based on specific molecules identified and properly characterized. I think this review may be attractive to readers interested in naturally occurring bioactive compounds, phytochemicals, ecology, and natural medicine. These fields are included in Aims and Scope de la journal. I think that the review contributes to consolidate and update the knowledge about D. graveolens (L.) Greuter.

In reference to English, the style is appropriate and understandable.

In reference to specific changes, the following errors have been detected:

  • Abstract: GREUTER appears in capital letters. Switch to Greuter.
  • Figure 1: Increase the font size for easier reading.
  • Section 2 “Phytochemistry”: Soxhlet must be capitalized with the first letter.
  • Figure 2: remove the reflection effect of the letters, make them difficult to read.
  • Section 2 "Phytochemistry": there is a missing space before the "On" in the following text: "Stara Planina mountain in Serbia [26].On the other hand ..."
  • Section 3 “Biological activities”: “Dittricchia graveolens” should be changed to “D. graveolens ".
  • Table 1: Ilicic acid entry, fourth column, appears "Fresh leaves [14] [14] [14] [14] [14] [14] [14]". Remove [14] [14] [14] [14] [14] [14] [14].
  • There is an error in the numbering of the subsections of section 3. The first subsection, “Antioxidant activity”, should be 3.1 and not 3.2.
  • Section 3.1: There is a missing space between “EC50” and the “=” symbol in “ABTS (EC50 = 7.58 ± 0.4 μg/mL)”.
  • Section 3.1: A comma (,) is missing between “mg/mL” and “lower” in “IC50 value of 7.7 mg/mL lower than the extracts…”
  • Section 3.2: There is a comma (,) missing between “cancer” and “and” in “several diseases, including cancer and that it is…”
  • Section 3.2: I think the cell lines they refer to should be specified in "1.8 μg / mL on investigated cell lines [34]"
  • Section 3.3: change “Plant’ metabolites” to “Plant metabolites”.
  • Section 3.3: Remove a space before “water” in “the antibacterial activity of the acetone/ water (4:1)…”
  • Section 3.5: Kojic acid in “comparison with Kojic acid as reference standard…” should not be capitalized.
  • Section 4 "Toxicology": rephrase "This is a good result since" to avoid giving personal judgments.
  • Section 5 "Conclusions": "Dittricchia graveolens" should be changed to "D. graveolens ".
  • References: Reference number 8 has two endpoints. Delete one.

Author Response

We thanks the reviewer for appreciative comment. 

We re-organize the Phytochemistry section and made all minor changes.

Reviewer 2 Report

This review under the title "Dittrichia graveolens (L.) Greuter, a rapidly spreading invasive plant: chemistry and bioactivity", can be accepted after revising the following comments:

  • An abbreviations and Methods sections should be added.
  • The synonyms of Dittrichia graveolens and the family of the plant should be added to the introduction with its biological and chemical importance.
  • It is better to summarise the biological effects of the plant under investigation in a table including its mode of administration and dosage form (or part) used

Author Response

We thanks the reviewer for the useful suggestions. A table has been added. 

Reviewer 3 Report

The paper " Dittrichia graveolens (L.) Greuter, a rapidly spreading invasive
plant: chemistry and bioactivity" give an updated and detailed overview on the chemical composition, biological activities, toxicology and the expansion of  D. graveolens plant. The manuscript was prepared carefully. This review can be accepted with minor revisions.
1. Check the typographical errors in Page 2, 3,5, 8, 16, 18, 19, 25.  I have noted and the words have been highlighted in yellow.

2.  The sentence "Bornyl acetate (41) was also present in the essential oil of the wild-growing Greek species collected at the full-flowering stage and investigated by Petropoulou et al., together with a lower percentage of epi-α-cadinol (116) (25.4% and 30.2%, respectively)" can be modified as "Bornyl acetate (41) with a lower percentage (25.4%) was also present in the essential oil of the wild-growing Greek species collected at the full-flowering stage and investigated by Petropoulou et al., together  epi-α-cadinol (116) (30.2%)"

3. Check the reference 38.

4. In the abstract, 52 references was used but in section (material and methods) they mentioned 51 articles.

5. In the Figure 3, I don't understand that there are red mark (X) in the arrow of ROS and antimicrobial activity.

Author Response

Corrections has been done:

  1. Typographical errors in Page 2, 3, 5, 8, 16, 18, 19, 25.
  2. The sentence "Bornyl acetate (41) was also present in the essential oil of the wild-growing Greek species collected at the full-flowering stage and investigated by Petropoulou et al., together with a lower percentage of epi-α-cadinol (116) (25.4% and 30.2%, respectively)" has been modified as "Bornyl acetate (41) with a lower percentage (25.4%) was also present in the essential oil of the wild-growing Greek species collected at the full-flowering stage and investigated by Petropoulou et al., together  epi-α-cadinol (116) (30.2%)"
  3. Reference 38 has been corrected.
  4. All references have been checked.
  5. Figure 3 has been modified.

Thanks 
